# Structural Quantile Normalization: a general, differentiable feature scaling method balancing Gaussian approximation and structural preservation

## Abstract

Feature scaling is an essential practice in modern machine learning, both as a preprocessing step and as an integral part of model architectures, such as batch and layer normalization in artificial neural networks. Its primary goal is to align feature scales, preventing larger-valued features from dominating model learning—especially in algorithms utilizing distance metrics, gradient-based optimization, and regularization. Additionally, many algorithms benefit from or require input data approximating a standard Gaussian distribution, establishing "Gaussianization" as an additional objective. Lastly, an ideal scaling method should be general, as in applicable to any input distribution, and differentiable to facilitate seamless integration into gradient-optimized models. Although differentiable and general, traditional linear methods, such as standardization and min-max scaling, cannot reshape distributions relative to scale and offset. On the other hand, existing nonlinear methods, although more effective at Gaussianizing data, either lack general applicability (e.g., power transformations) or introduce excessive distortions that can obscure intrinsic data patterns (e.g., quantile normalization). Present non-linear methods are also not differentiable. We introduce Structural Quantile Normalization (SQN), a general and differentiable scaling method, that enables balancing Gaussian approximation with structural preservation. We also introduce Fast-SQN; a more performance-efficient variant with the same properties. We show that SQN is a generalized augmentation of standardization and quantile normalization. Using the real-world "California Housing" dataset, we demonstrate that Fast-SQN outperforms state-of-the-art methods—including classical and ordered quantile normalization, and Box-Cox, and Yeo-Johnson transformations—across key metrics (i.e., RMSE, MAE, MdAE) when used for preprocessing. Finally, we show our approach transformation differentiability and compatibility with gradient-based optimization using the real-world "Gas Turbine Emission" dataset and propose a methodology for integration into deep networks.

## 1 Introduction

Feature scaling is an essential step in the machine learning pipeline, crucial for optimizing model performance and ensuring reliable outcomes. Many machine learning algorithms, particularly those that rely on distance metrics, gradient-based optimization, or regularization, are sensitive to the scale of input features. Without appropriate scaling, features with larger numeric ranges can dominate the learning process, resulting in biased models, slower convergence, and decreased predictive accuracy. Effective feature scaling addresses these issues by balancing the contribution of all features, thereby enhancing the stability and efficiency of the learning process. Consequently, feature scaling is widely used as a preprocessing step to prepare data before model training. Moreover, in many modern deep learning architectures, scaling has become an integral part of the learning process itself. For instance, techniques like batch-, weight-, and layer normalization embed feature scaling into neural network layers to reduce internal covariate shift, stabilize training, and accelerate convergence (Xu et al., 2019; Liu et al., 2020; Salimans & Kingma, 2016; Ren et al., 2017).

In addition to harmonizing the scale of features, many machine learning algorithms either require or significantly benefit from input data that approximates a standard Gaussian distribution. Gaussian distribution has desirable properties, such as symmetry and defined statistical characteristics, that can enhance the performance of algorithms by improving the effectiveness of optimization techniques and statistical inference (Roy, 2003). Data that closely resembles a Gaussian distribution can lead to faster convergence rates and more reliable model outcomes. For example, k-means clustering, linear regression, and neural networks often assume or perform better with normally distributed data, resulting in more accurate parameter estimates, improved predictions, and enhanced convergence rates (Napoleon & Lakshmi, 2010; Li et al., 2012; Farokhnia & Niaki, 2020; Wang et al., 2017). Importantly, existing research has established that making data more Gaussian is not the only important objective when considering feature scaling. It is equally important to scale the data such that the inherent structure and relationships among features are preserved, avoiding excessive distortion that could result in valuable information loss (Peng et al., 2007). Furthermore, it is beneficial for such approaches to be differentiable and enable gradient-based optimization straightforwardly.

Traditionally, various linear scaling methods, such as rescaling, mean normalization, standardization (STD), robust scaling, and unit vector normalization, have been used to adjust the range and spread of input data (Ali et al., 2014). Some of these methods transform data to have a mean of zero and a standard deviation of one, which can help approximate Gaussian-like behavior. In addition, such methods are differentiable and, hence, they are commonly employed as integral parts in machine learning models that rely on gradient-based optimization (Xu et al., 2019). However, being inherently linear, these methods are unable to change a distribution relative to scale and offset. Consequently, they fail to effectively combat nonlinear deformations in a feature's underlying scale, such as if the data was measured on an exponential scale (Kao, 2010).

To address these limitations, various non-linear methods that transform data into a form that more closely resembles a Gaussian distribution have found their way into machine learning methodology. Some of the oldest popularly applied "Gaussianizing" transformation methods started emerging in the late 20th century, as the family of "power transformations." These can be characterized as methods that apply combinations of relatively simple functions, such as exponentials or logarithms, with the aim to increase Gaussian resemblance when applied to certain commonly found distribution types. Most prominently, the power transformations include the Box-Cox transformation (BXC), which works very well for un-skewing log-normal distributions (Box & Cox, 1964), as well as the Yeo-Johnson transformation (YJN), which is a more flexible alteration of the latter, namely in being able to handle negative values (Yeo & Johnson, 2000). As a consequence of the simple functions they consist of, these transformations typically only introduce a modest amount of curvature and distortion to their input domain, which is reflected by their general ability to preserve inherent structure and internal relationships in its subject data. However, their application in a *general* context, where the target distribution is not known in advance, is problematic as applying such a transformation on a distribution unlike its targeted distribution(s) often does not yield results that come close to a Gaussian distribution at all. Additionally, these transformations are not fully differentiable, hindering their application within pipeline architectures that require gradient-based optimization.

Apart from power transformations, the other prominent approach to "Gaussianizing" data is quantile normalization, which we, henceforth, address as Classical Quantile Normalization (CQN), along with a recent variation, which is known as Ordered Quantile Normalization (OQN) (Peterson & Cavanaugh, 2019). Quantile normalization is the most direct "Gaussianizing" technique in that it redistributes its input to fit a perfect Gaussian distribution. However, since the induced output space of perfect Gaussian distributions is very small, a severe information loss is inevitable during the transformation (Peng et al., 2007). This is the main drawback of using quantile normalization, as removing substantial information from the original vector results in models trained on the transformed data potentially not being able to capture all relevant patterns in the underlying phenomena. It is for this reason that power transformations remain a more popular choice despite the generality of quantile normalization techniques. Additionally, quantile normalization is also not a differentiable operation, making it equally unfit for direct integration into gradient-optimized architectures.

Against this background, we introduce Structural Quantile Normalization (SQN), a novel, differentiable, and general feature scaling transformation that balances Gaussian resemblance and structural preservation. The core principle of SQN relies on differentiating between, what we call, "global" and "local" distribution of a feature. The former refers to globally prevalent distribution trends, while the latter captures finer, neighborhood-level relationships among the individual data points

themselves. SQN's objective is to, as such, "Gaussianize" the data "global distribution" while maintaining its "local distribution." To this end, SQN builds on kernel density estimation, utilizing kernel density estimation to capture the "global distribution" while sparing "local distribution,". We also propose Fast-SQN; a faster variant that relies on cubic spline interpolation to improve computational efficiency while retaining SQN advantages. Furthermore, we prove that in the limit cases of SQN's inherent transformation parameter, the transformation approaches the behaviors of STD and CQN, respectively. In this sense, SQN can be seen as a generalized augmentation of these, in essence, antipodal, feature scaling approaches, which enables SQN to achieve a favorable balance between the benefits and drawbacks of STD and CQN. We compare Fast-SQN to traditional standardization (STD), state-of-the-art power transformations (i.e., BXC, YJN), and state-of-the-art quantile normalization techniques (i.e., CQN, OQN) when used as feature scalers in real-world regression models. We do so by inspecting their impact on key performance metrics (i.e., RMSE, MAE and MdAE) and visually inspecting the residual distributions, providing key insights. We train and evaluate the performance of five neural network architectures of various sizes on the California Housing Dataset (Pace & Barry, 1997) to asses the scalability of Fast-SQN. Our findings show that Fast-SQN outperforms existing normalization methods in all considered metrics. In the context of this work, we demonstrate the effectiveness of Fast-SQN within a pipeline utilizing gradient descent optimization on the Gas Turbine Emission dataset (Kaya et al., 2019). By doing so, we highlight Fast-SQN's suitability for integration into machine-learning workflows that rely on gradient-based optimization.

In more detail, our main contributions are the following:

- We propose a novel, general and differentiable feature scaling technique, called SQN, that balances Gaussian resemblance and structural preservation, along with a computationally faster variant, called Fast-SQN,
- We show that SQN is a generalization of both STD and CQN, approximating their respective behavior in the limit cases of its transformation parameter.
- We evaluate Fast-SQN using real-world data against several popular state-of-the-art feature scaling methods, namely STD, CQN, OQN, BXC, and YJN, considering machine learning models of different sizes.
- We show that Fast-SQN is superior in terms of RMSE, MAE, MdAE and provide insights regarding the residual distribution.
- We show Fast-SQN's transformation differentiability and propose a methodology of integrating our method into deep neural networks.

In Section 2, we review the related work, focusing on quantile normalization, the original methodology on which ours builds, and other associated techniques used by our new method. In Section 3, we present our approach, offering a detailed explanation of the implementation, a high-level overview and an intuitive perspective. There, we also discuss the transformation parameters and their impact on transformation behavior and performance. Section 4 discusses our case study and the evaluation result. There, we also demonstrate the differentiability of SQN, along with its compatibility with gradient-based optimization. Section 5 concludes the work.

## 2 RELATED WORK

Here we review related work, focusing on CQN—the original quantile normalization method on which ours builds—and other techniques used by our new method, in particular, kernel density estimation and PCHIP monotonic spline interpolation. We also discuss OQN, and the BXC and YJN power transformations, all of which are benchmark methods in our evaluation.

### 2.1 QUANTILE NORMALIZATION

In this work, with the term *quantile normalization* we refer to the general family of feature scaling techniques that apply a rank-based mapping to force its input data into a Gaussian shape, guaranteeing effectiveness on any input distribution. As such, quantile normalization can be considered the prototype for generally applicable Gaussianizing transformations.

**Classical Quantile Normalization (CQN)** CQN is the forerunner of quantile normalization. Given an input vector of numeric data with $n$ distinct values, CQN executes the following steps:

1. Sort the input vector in ascending order.

2. Map the sorted input vector's $i^{th}$ distinct value to the $(\frac{i}{n})^{th}$ Gaussian quantile.

The derived mapping is a piece-wise one-to-one function that transforms the input data into a Gaussian one. The sorted index of a vector entry is commonly denoted as its "rank", which is why the described procedure is called a "rank mapping". This corresponds to the idea of a "quantile" within the vector, which gives rise to the name discrete "quantile normalization". The transformation is perfect in case of no duplicates in the input data, i.e., when the size of the vector is $n$. However, even if this assumption does not hold, the output will still be as close to Gaussian as possible (Peterson & Cavanaugh, 2019), and the deviations are almost negligible with small tie-to-total data ratios.

An important shortcoming of CQN is that the domain of the derived function is the input data themselves; the function is not defined outside this domain. Hence, the applicability of CQN in machine learning tasks is limited. A commonly employed workaround today is to add linear interpolation and clip values that are out of range to the range limits.

An additional shortcoming, imposed by the described rank mapping, is that it almost completely decouples the input vector's entries from their context, leaving their relative order as the only information that is preserved (Peng et al., 2007). As such, the counter-distortion that is applied to force the data to fit a Gaussian shape tends to wash out the inherent structure and relationships between the data points, negatively impacting model performance in turn. In addition, the rank mapping is inherently discrete and discontinuous, making it unfit for differentiation.

**Ordered Quantile Normalization (OQN)**   OQN is a modern adaptation of CQN, notably improving on the original method in two ways. First, it includes linear interpolation and extrapolation to extend the domain beyond the input values, providing improved applicability to general machine learning tasks. Second, CQN utilizes an interpolated lower-density "sample" of the Gaussian quantiles to reduce runtime overhead in scenarios with large datasets. While this measure results in slight inaccuracies in the output, these are often negligible. (Peterson & Cavanaugh, 2019).

Despite these improvements, OQN tends to wash out the inherent structure and relationships between the data points similarly to CQN, which, as discussed, negatively impacts its performance (Peng et al., 2007). Finally, CQN is also non-differentiable.

## 2.2   POWER TRANSFORMATIONS

In this work, the term *power transformations* refers to the family of feature scaling techniques which apply combinations of relatively simple functions with the aim to increase Gaussian resemblance when applied to certain commonly found distribution types. Most prominently, the power transformations include the Box-Cox transformation (BXC), which works very well for un-skewing log-normal distributions (Box & Cox, 1964). The other prominent power transformation is the Yeo-Johnson transformation (YJN), which is a more flexible alteration of the latter, namely in being able to handle negative values (Yeo & Johnson, 2000). Both of these methods are dependent on an internal parameter $\lambda$. The $\lambda$ parameter controls the degree of distortion performed by the transformation in an internally used exponential function. Its value, however, is not directly customizable but is set by a hidden optimization procedure in favor of the resulting Gaussian resemblance. It is due to the discrete nature of this optimization step that the power transformations are not differentiable.

## 2.3   (GAUSSIAN) KERNEL DENSITY ESTIMATION [**KDE**]

Kernel Density Estimation (KDE) is a statistical technique for producing a smooth estimate of a distribution's *probability density function* (pdf), based on a discrete sample of that distribution. Its core idea is to place a non-negative, spike-formed kernel function centered at each data point and construct the estimated pdf by additively accumulating the kernels (Węglarczyk, 2018). Gaussian KDE uses a Gaussian curve as its kernel, producing smooth and continuous estimations (Figure 1).

The width of the kernels, corresponding to the spread of the underlying bell curves, is a parameter of the KDE technique that influences the extend to which patterns of smaller size are smoothed out in favor of leaving only fundamental global distribution patterns when the kernel width is large enough.

Apart from the differentiability of KDE, this property is of particular importance for its utilization in SQN, as is subsequently described in Section 3.

Figure 2 shows a realistic sample distribution's Gaussian KDE overlaid on its histogram. The KDE visibly captures only the general distribution trend while omitting the local structure in this instance.

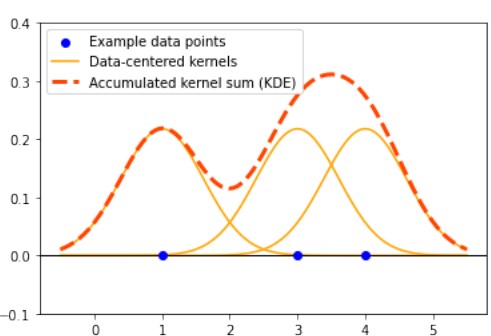 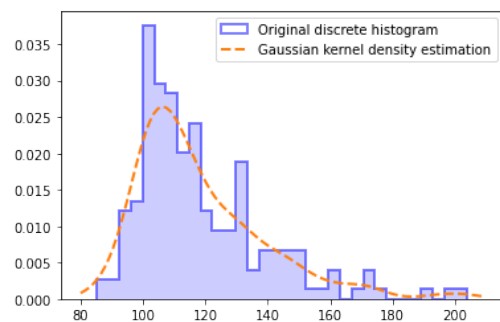

Figure 1: Accumulating Gaussian kernels      Figure 2: A sample distribution's KDE

### 2.4 THE **PCHIP** MONOTONIC SPLINE INTERPOLATION ALGORITHM

Another technique used by SQN is the PCHIP algorithm. PCHIP achieves *monotonic* cubic spline construction (Fritsch & Butland, 1984) in the sense that the resulting interpolation function is guaranteed to be monotonic as long as the input anchor points are monotonically increasing or decreasing. This is crucial for the interpolation to be bijective, and, thus, invertible, which is a critical requirement of this work as further discussed in Section 3. In addition to monotony, the PCHIP algorithm provides smoothness and differentiability in its resulting interpolation. Due to these reasons, the PCHIP monotonic spline interpolation algorithm is utilized in this work.

Given a list of input-output anchor points, PCHIP constructs a piece-wise cubic interpolation, where a separate spline segment is fit between each pair of adjacent anchor points. To ensure that the fused spline is continuous and differentiable, the cubic segments are constructed to match the anchor points' values at their boundaries, as well as match the slope of the adjacent segment at the boundary. PCHIP computes the anchor point slopes as the harmonic mean of adjacent differences, which is a continuously differentiable operation in terms of the input anchor points' coordinates.

## 3 INTRODUCING STRUCTURAL QUANTILE NORMALIZATION

We propose Structural Quantile Normalization (SQN), a feature scaling method that: (i) is general and effectively applicable to any numeric feature, regardless of its distribution, (ii) transforms its input feature to more closely resemble a Gaussian distribution, (iii) balances the preservation of local structure against counter-distortion introduced in favor of Gaussianization, based on a single parameter $\sigma$, and (iv) is differentiable with respect to all of its inputs.

SQN achieves the above by utilizing the *continuous analog* of the discrete rank mapping technique used by quantile normalization. In particular, a *continuous* distribution $a$ may be mapped onto another distribution $b$ by matching values of equal *quantile*, which is given by each distribution's cumulative distribution function (cdf). Formally, with ppf denoting the percent point function (inverse of the cdf), the quantile mapping looks like $x \mapsto cdf_b^{-1}(cdf_a(x))$, or, equivalently, $x \mapsto ppf_b(cdf_a(x))$.

Building on this foundation, SQN follows a three-step procedure:

1. Use Gaussian KDE to construct a smooth probability density function (pdf) on the input vector $\boldsymbol{v}$, denoted $\text{kde}_{\boldsymbol{v}}(x)$, which is globally defined and continuous.

2. Obtain the corresponding smoothed cdf, denoted $cdf_{\boldsymbol{v}}^*(x)$, by integrating $\text{kde}_{\boldsymbol{v}}$, via $cdf_{\boldsymbol{v}}^*(x) = \int_{-\infty}^{x} \text{kde}_{\boldsymbol{v}}(t)dt$.

3. Apply the continuous quantile mapping from $\text{kde}_{\boldsymbol{v}}(x)$ to the standard Gaussian distribution on input $x$, such that $SQN(x|\boldsymbol{v}) = \Phi^{-1}(cdf_{\boldsymbol{v}}^*(x))$ where $\Phi^{-1}$ denotes the Gaussian ppf.

Step 1 results in a continuous and smooth approximation of the original vector $v$'s distribution, which is attained in a differentiable manner. Furthermore, it ensures that all values—even those not present in the original input vector $v$—are included in the domain of the transformation. The use of Gaussian KDE for constructing the pdf notably makes for a second, yet completely separate, appearance of the Gaussian bell curve in our method. Its distinctive properties, particularly differentiability as well as its "smoothing effect" in the pdf estimation, make Gaussian KDE a very natural choice of component for SQN to use here. Step 2 results in a continuous cdf that approximately fits the discrete input vector's distribution, based on which the continuous quantile mapping can be used in a differentiable manner. Step 3 completes the method by transforming the input vector's domain such that the approximated pdf is mapped onto a standard Gaussian distribution. This guarantees a certain level of Gaussian resemblance in the output no matter the original distribution.

While enabling the use of the fully differentiable quantile mapping on the one hand, the Gaussian KDE component additionally results in the very effect being achieved that is needed for structural preservation. Specifically, the smoothing effect in Gaussian KDE results in merely the globally dominating distribution trend being captured in the resulting pdf, while finer, more local data patterns are virtually undepicted in the pdf. Unlike in CQN, where the entire original distribution is "undone" to be mapped to a Gaussian, applying the continuous quantile mapping from the pdf obtained by KDE onto a Gaussian thereby only undoes the dominating distribution trend, which was captured by KDE, while preserving local structure, which was expunged by the latter.

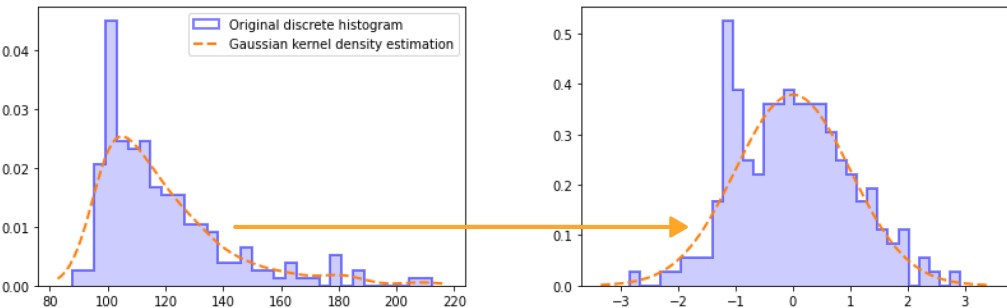

Figure 3: Under the hood of the SQN transformation

Figure 3 shows an intuitive visualization of how the SQN transformation operates, using the smooth KDE as an estimated "global distribution" which is mapped to Gaussian shape, leaving local patterns intact. In this example, the transformation removes the left skew predominant in the original distribution while preserving local structure that was present in the input vector. The irregular value spike around the '100' tick in the original distribution in the left image is a good example for 'local' structure which deviates from the overall distribution trend, which appears to resemble a log-normal distribution. The right image shows that the application of the SQN transformation does not distort the spike to further fit the Gaussian shape, but preserves it as the spike around the '-1' tick.

### 3.1 THE KERNEL WIDTH PARAMETER $\sigma$ AND ITS LIMIT CASES

Gaussian KDE is parameterized by the width of the kernels, and this parameter is consequently inherited by SQN, denoted $\sigma$ henceforth. As described in Section 2.3 on Gaussian KDE, as the kernel width expands, the approximated pdf becomes smoother and deviates further from the actual distribution, capturing only broader, more global distribution patterns. Conversely, as the kernel width decreases, the approximated pdf becomes more jagged and closely aligns with the actual distribution, capturing finer, more local distribution patterns. Importantly, since SQN transforms the KDE onto a perfect Gaussian, causing the loss of the structure captured by KDE as occurs in CQN, the structure that is preserved by SQN is crucially that which is *not* captured by KDE and thus not precisely mapped to Gaussian shape but preserved relative to the approximated distribution.

As highlighted in the introduction, SQN is a generalized augmentation of STD and CQN, in that the transformation approximates their respective behavior in the limit cases of the parameter $\sigma$. In particular, a very low value for $\sigma$ leads to the input vector's distribution being captured in its entirety and perfectly mapped to Gaussian shape, resulting in behavior identical to CQN. A proof for this

proposition can be found in appendix A. On the other hand, a very high value for $\sigma$ causes the KDE to converge to a single Gaussian centered at the input vector's arithmetic mean. Since mapping this Gaussian onto the standard Gaussian is a linear transformation, which moreover maps the input vector's mean to zero, SQN becomes directly proportional to STD under these circumstances. A proof of this proposition can likewise be found in appendix A.

### 3.2 EXTENDING THE METHOD WITH FAST-SQN

The principal SQN method as described thus far presents two drawbacks when it comes to its computational performance. Firstly, evaluating $cdf_v^*$ on the vector $\boldsymbol{v}$ itself as done during the operation is a $O(n^2)$ operation, as the kde is influenced by every single entry. This makes it considerably slower than the state-of-the art transformations, the majority of which are light-weight and run with $O(n)$ time complexity. Additionally, the operation of building the integrated kde from the input vector cannot be inverted easily, which makes the implementing the inverse transformation nontrivial.

In light of these capacities for improvement, we propose a performance-accelerated extension of SQN, namely Fast-SQN, which simultaneously addresses both mentioned performance drawbacks while sacrificing some accuracy relative to native SQN. In particular, in order to avoid having to evaluate the computationally intensive KDE at every entry of the input to be normalized, Fast-SQN adds a spline interpolation layer to the pdf computation, where only an evenly spaced sub-sample of the input values are directly passed through KDE, while a cubic spline is fit on the resulting values over the entire domain. As such, this procedure gives rise to a second transformation parameter being added to Fast SQN, which is the number of anchor points that are directly sampled through KDE, denoted henceforth as $s$. This parameter balances the accauracy of the approximation with performance, as discussed in more detain in section 4, using our case study dataset. To cover values beyond the input vector's domain, the spline is extrapolated with a linear extension, matching the spline's slope at each end, in a fashion resembling the extrapolation mechanism used in OQN. Conveniently, the inherent smoothness of the KDE makes it suitable for interpolation on a relatively small number of directly evaluated representative anchor points. For the task of calculating appropriate slopes at the anchor points, we make use of the PCHIP algorithm, designed to guarantee the monotony of the interpolates spline, a crucial property in upholding bijectivity in the transformation. Compared to native SQN, Fast SQN has the great advantage that (i) the runtime complexity is reduced to $O(n)$ and (ii) the inversion of the spline becomes trivial via the cubic formula, all while sacrificing little accuracy in the transformation. Ultimately, Fast-SQN retains all the desired properties of SQN that were discussed in the previous section, and should hence be chosen for practical application due to its strongly reduced computational footprint. For a fair comparison to other light-weight scaling methods, we therefore exclusively use Fast-SQN during the evaluation process of this paper.

## 4 EVALUATION

Here, we evaluate the real-world performance impact of training regression models on features transformed using Fast-SQN, and compare it to the state-of-the-art methods: classical standardization (STD), quantile normalization (CQN), ordered quantile normalization (OQN) Peterson & Cavanaugh (2019), Box-Cox power transform (BXC) Box & Cox (1964), and Yeo-Johnson power transform (YJN) Yeo & Johnson (2000). We also provide insights on how Fast-SQN compares to SQN and demonstrate the compatibility of our approach with gradient-based optimization.

### 4.1 EVALUATION CASE STUDY

For maximum relevance to current practices in regression model design, we train artificial neural networks (ANNs) using the "California Housing" dataset (Pace & Barry, 1997), a popular choice for regression model evaluation. Industry-deployed ANNs vary in layer sizes depending on many factors, such as available computation resources. Therefore, we test our method on ANNs with different layer shapes, progressing from a small, shallow model to a large, deeper model. Doing so gives us additional insight on how the effectiveness of SQN and other methods changes as the models capture increasingly complex patterns within the data.

All models are fully connected feed-forward ANNs using the ReLu activation function for all hidden layers and the identity function for the output layer. They are trained using the Adam opti-

mizer at a learning rate of 0.001, each for 1000 iterations using the same machine (Apple M1/8 cores/3.2GHz/16GB). We used a random 80-20 dataset split for training and testing. All benchmark scalers are non-parametric. We set the parameters for Fast-SQN to $\sigma = 0.2$ and $s = 16$, as found by a grid search approach to strike an appropriate balance between quality and performance.

The ANNs trained are: (i) **Model 1**: A relatively small ANN of the shape (8, 20, 1), representing a regress that prioritizes computational efficiency; (ii) **Model 2** An ANN with the layer sizes (8, 40, 20, 1), to make for a medium-sized model, representing a typical architecture for the dataset use case; (iii) **Model 3** A larger ANN with the layer sizes (8, 90, 45, 1), makes for a model capable of capturing a greater amount of patterns and relationships within the training dataset; (iv) **Model 4** A comparatively deep ANN with layer sizes (8, 40, 30, 20, 10, 1); and (v) **Model 5** An even deeper ANN with the layer sizes (8, 20, 20, 20, 10, 10, 10, 1), to make for a model capable of capturing more elaborate patterns and relationships within the training dataset.

We use each competing feature scaler to normalize the target column and compare the performance of the resulting models after transforming their predictions back to the original scale. We use multiple error metrics to assess model performance for a holistic perspective, namely Root Mean Squared Error (RMSE), Mean Absolute Error (MAE) and Median Absolute Error (MdAE).

Table 1: Evaluated error metrics for five different neural network shapes

| Model (architecture) | Metric | Feature scaler | | | | | |
|---|---|---|---|---|---|---|---|
| | | STD | CQN | OQN | BXC | YJN | Fast-SQN |
| Model 1 (8, 20, 1) | RMSE | 0.568 | 0.630 | 0.620 | 0.569 | 0.572 | **0.558** |
| | MAE | 0.392 | 0.398 | 0.397 | 0.377 | 0.377 | **0.367** |
| | MdAE | 0.271 | 0.244 | 0.243 | 0.251 | 0.245 | **0.236** |
| Model 2 (8, 40, 20, 1) | RMSE | 0.528 | 0.579 | 0.624 | 0.546 | 0.525 | **0.521** |
| | MAE | 0.369 | 0.367 | 0.391 | 0.357 | 0.349 | **0.337** |
| | MdAE | 0.264 | 0.223 | 0.232 | 0.223 | 0.232 | **0.206** |
| **Model 3** (8, 90 45, 1) | RMSE | 0.521 | 0.585 | 0.567 | 0.514 | 0.509 | **0.508** |
| | MAE | 0.348 | 0.369 | 0.357 | 0.334 | 0.333 | **0.325** |
| | MdAE | 0.226 | 0.214 | 0.208 | 0.212 | 0.210 | **0.199** |
| Model 4 (8, 40, 30, 20, 10, 1) | RMSE | 0.521 | 0.596 | 0.587 | 0.531 | 0.516 | **0.515** |
| | MAE | 0.339 | 0.370 | 0.367 | 0.348 | **0.334** | **0.334** |
| | MdAE | 0.216 | 0.224 | 0.216 | 0.220 | **0.206** | **0.206** |
| Model 5 (8, 20, 20, 20, 10, 10, 10, 1) | RMSE | 0.537 | 0.597 | 0.596 | 0.543 | 0.542 | **0.522** |
| | MAE | 0.362 | 0.383 | 0.388 | 0.347 | 0.367 | **0.339** |
| | MdAE | 0.237 | 0.230 | 0.250 | **0.209** | 0.242 | **0.209** |

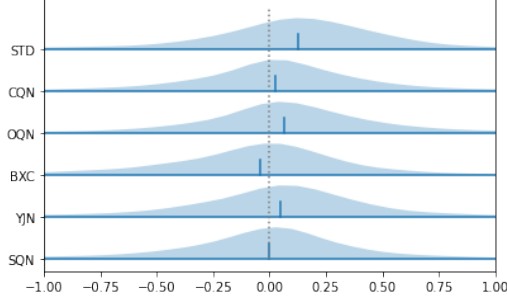 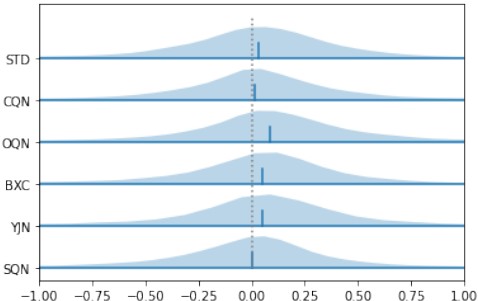

Figure 4: Residual distributions for model 2    Figure 5: Residual distributions for model 3

## 4.2    EVALUATION RESULTS

The evaluation error metric results are collectively presented in Table 1. The results indicated in bold indicate the best-achieving ones. Fast-SQN achieves the best performance in all metrics and

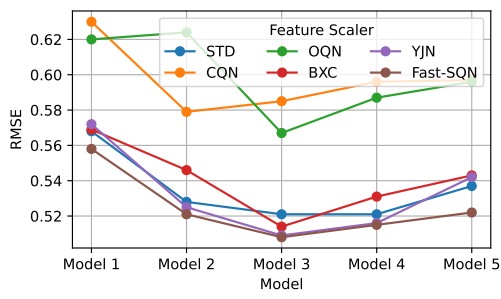 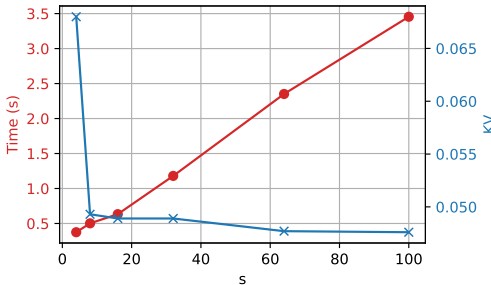

Figure 6: RMSE (all models and scalers)          Figure 7: Fast-SQN vs SQN

all models considered, tying only with BXC and YJN in models 4 and models 5, respectively, in the metrics MAE and MdAE.

Notably, the power transformations BXC and YJN perform better compared to CQN and OQN in all metrics and across all models considered—as also shown in figure 6. Arguably, the power transformations seem to fit the dataset distribution well, going by their good performance. The generality of CQN and OQN comes at the clearly observable cost of performance deterioration in this case study. Nevertheless, Fast-SQN performs better than both BXC and YJN, demonstrating the improvement which balancing structural preservation can achieve, even when building on the same philosophy as CQN and OQN.

Interestingly, STD is competitive to power transformations and SQN, demonstrating the robustness of STD and justifying its widespread usage—Figure 6. Nevertheless, SQN performs better than STD across all models considered. While STD achieves its best $RMSE$ score at 0.521, SQN achieves an error below this benchmark in models 2, 3, and 4, while reaching its optimum at 0.508, marking an improvement of 2.3%. The distinction is larger with the $MAE$ and $MdAE$ metrics, where SQN achieves minimum errors 4.1% and 5.0% below those of STD, respectively. Still, STD's competitiveness in these results suggests that a higher Fast-SQN $\sigma$ choice (leaning more towards STD compared to CQN) can be a better choice for this dataset.

With model 3 (8, 90, 45, 1), Fast-SQN achieves the overall best result in the RMSE metric and very competitive results in the other metrics. This is also reflected in Figure 6. To verify the statistical significance of our testing results, we performed the ANOVA test and follow-up pairwise t-tests on the residuals for Model 3. With an ANOVA p-value of around $2.4 \cdot 10^{-96}$ and all pairwise t-test p-values below $0.01$, we affirm the validity of our conclusions.

Finally, to shed more light on the transformation properties of SQN, we visually inspect the residual distributions. Figures 4 and 5 illustrate the residual distribution for model 2 and 3, respectively. Both figures convey how transforming the input data's "global distribution" into Gaussian shape is effective in minimizing the resulting models' bias as its residual distribution is centered closest to 0. This observation reflects what existing research has shown, in that bias in neural networks tends to be less pronounced if its input data generally follows a Gaussian distribution (Wang et al., 2017), while also reaffirming the favorable effect of an increased structural preservation when compared to CQN and OQN. Ultimately, the residual distribution achieved through the use of SQN is a conveyance of the healthy balance which the method is able to achieve.

### 4.3 FAST-SQN VS SQN AND THE EFFECT OF THE $s$ PARAMETER

The parameter $s$ of Fast-SQN balances approximation of SQN-behavior and computational time. Figure 7 shows the runtime for various choices for $s$ (i.e., 4, 8, 16, 32, 64, and 100) for transforming the target column of the California Housing dataset and applying the inverse transformation. It also shows the divergence between the resulting distributions of Fast-SQN and SQN, obtained through the Kolmogorov-Smirnov (KV) agnostic. Fast-SQN achieves great SQN resemblance even for small $s$ values with a sweet spot at 8 or 16 points—runtime increases greatly for little KV gains henceforth.

### 4.4 Gradient-based Optimization with SQN

In order to demonstrate Fast-SQN's practical differentiability along with its compatibility with gradient-based optimization methods, we run two iterative optimization epochs, acting on the value to be transformed and on the input base vector, respectively. First, we test optimizing the transformation input value $x$ to meet a sample target condition after being transformed by Fast-SQN. As the sample feature vector for testing, we use the *AFDP* column in the *Gas-Turbine CO and NOx Emission Data* dataset (mis, 2019), which we once again call $v$. As our target condition for this test run, we arbitrarily select $SQN(x|v) = -2.0$. After setting $x = 4.2$ as $x$'s initial state, we now iteratively offset $x$ using a basic Stochastic Gradient Descant (SGD) optimizer on the loss metric $(SQN(x|v) - (-2))^2$, in correspondence with our previously selected target condition. As shown in Figure 8, $x$ converges to a lower value at around 3.5, which approximately satisfies our target condition, after around 10 to 15 iterations at a constant learning rate of 0.1. In the next step, we demonstrate equal differentiability of SQN with regard to the base vector $v$, by having the SGD optimizer offset $v$ as its target tensor instead of $x$. With the final state of the first testing epoch becoming the new initial state, we arbitrarily choose $SQN(3.8|v) = -2.5$ as a new target condition for $v$ to be fitted to. After 15 to 20 iteration steps on the corresponding loss metric $(SQN(3.8|v) - (-2.5))^2$, Figure 9 shows that the entries in $v$ have been smoothly altered by the optimizer, so as to approximately fit our new target condition. These results demonstrate that the SQN transformation is fit to be used as a gradient-optimized component within processing pipelines for machine learning.

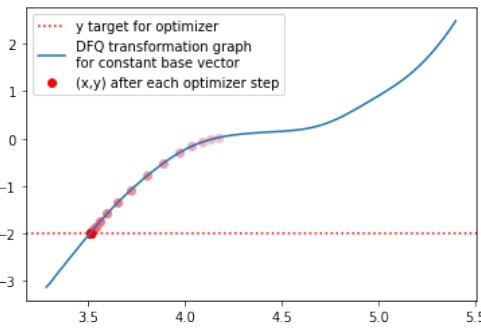

Figure 8: Gradient descent on $x$

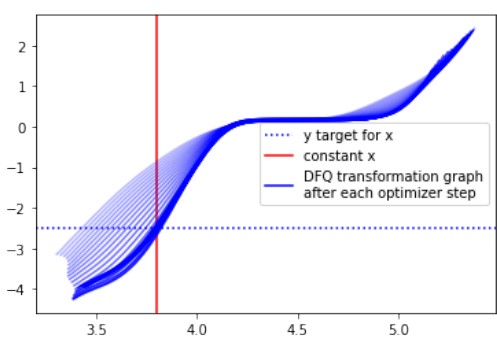

Figure 9: Gradient descent on $v$

## 5 Conclusion

We have found Structural Quantile Normalization, apart from being the first transformation of its kind to be fully differentiable, to also be competitive in terms of model performance impact. It strikes a healthy balance between Gaussianizing and maintaining inherent structure in its the data it transforms. We applied SQN on the *California Housing* dataset regression task and found superior performance when compared to training the same model using state-of-the art transformations.

As noted in the introduction, we envision the applicability of Fast-SQN not only as a preprocessing tool but as the first quantile batch normalization layer, as made possible by its differentiability. With its efficient spline computation model, it was created with this use in mind, given that performance becomes increasingly critical when iteration-wise transformation is required during model training. Future work could also compare and contrast other ways of facilitating the interpolation, possibly resulting in better performance or more retained accuracy than the spline-based method used in Fast-SQN. In addition, we envision the addition of differentiable heuristics that are capable of determining fitting values for the transformation parameters based on the feature vector, in order to move towards making Fast-SQN a non-parametric transformation. Ultimately, we hope that the development of differentiable and structure-preserving feature scaling techniques will continue to contribute to the creation of improved model architectures in the field of machine learning.

**Reproducibility Statement** To facilitate straightforward reproduction of the experimental results presented in this paper, we share source code for SQN itself as well as the Python script we used for model evaluation and comparison between the competing feature scaling methods in a supplementary document.

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

## A  PROOFS

**Proposition 1.** *As the KDE kernel width $\sigma$ goes to infinity, SQN approaches a linear transformation which maps the mean of the input vector to zero, making it proportional to the standardization transformation. Given the notations introduced in Section 3, we have*

$$\lim_{\sigma\to\infty} SQN_\sigma(x|\boldsymbol{v}) = \frac{x-\overline{\boldsymbol{v}}}{\sigma} \propto STD(x|\boldsymbol{v}) \tag{1}$$

*Proof.* As introduced in Section 3, SQN is defined for a vector $\boldsymbol{v}$ with $n$ entries, an input $x$ and a kernel width of $\sigma$ as

$$SQN_\sigma(x|\boldsymbol{v}) = \Phi^{-1}(cdf^*_{\boldsymbol{v},\sigma}(x)) \quad \text{where} \quad cdf^*_{\boldsymbol{v},\sigma}(x) = \frac{1}{n}\sum_{i=1}^{n}\Phi\left(\frac{x-\boldsymbol{v}_i}{\sigma}\right)$$

Since $\Phi$ is an analytic function, we can use its Taylor series to expand

$$\Phi(t) = \Phi(0) + \Phi'(0)t + \frac{1}{2}\Phi''(0)t^2 + \alpha(t)t^3$$

$$= \frac{1}{2} + \frac{1}{\sqrt{2\pi}}t + \alpha(t)t^3$$

where $\alpha$ is analytic, too. We know that the degree 2 term must be zero, since $\Phi$ is point-symmetric around $(0,\frac{1}{2})$. Similarly, we can expand $\Phi^{-1}$ around its center of point symetry at $(\frac{1}{2},0)$ as

$$\Phi^{-1}(t) = \Phi^{-1}(\tfrac{1}{2}) + (\Phi^{-1})'(\tfrac{1}{2})(t-\tfrac{1}{2}) + \tfrac{1}{2}(\Phi^{-1})''(\tfrac{1}{2})(t-\tfrac{1}{2})^2 + \beta(t-\tfrac{1}{2})(t-\tfrac{1}{2})^3$$

$$= \sqrt{2\pi}(t-\tfrac{1}{2}) + \beta(t-\tfrac{1}{2})(t-\tfrac{1}{2})^3$$

where $\beta$ is analogously analytic. Then we have

$$\Phi^{-1}(cdf^*_{\boldsymbol{v},\sigma}(x)) = \Phi^{-1}\left(\frac{1}{n}\sum_{i=1}^{n}\Phi\left(\frac{x-\boldsymbol{v}_i}{\sigma}\right)\right)$$

$$= \Phi^{-1}\left(\frac{1}{n}\sum_{i=1}^{n}\left(\frac{1}{2} + \frac{1}{\sqrt{2\pi}}\left(\frac{x-\boldsymbol{v}_i}{\sigma}\right) + \alpha\left(\frac{x-\boldsymbol{v}_i}{\sigma}\right)\left(\frac{x-\boldsymbol{v}_i}{\sigma}\right)^3\right)\right)$$

$$= \Phi^{-1}\left(\frac{1}{2} + \frac{1}{\sqrt{2\pi}}\left(\frac{x-\overline{\boldsymbol{v}}}{\sigma}\right) + \frac{1}{n}\sum_{i=1}^{n}\left(\alpha\left(\frac{x-\boldsymbol{v}_i}{\sigma}\right)\left(\frac{x-\boldsymbol{v}_i}{\sigma}\right)^3\right)\right)$$

$$= \Phi^{-1}\left(\frac{1}{2} + \frac{1}{\sqrt{2\pi}}\left(\frac{x-\overline{\boldsymbol{v}}}{\sigma}\right) + \frac{1}{n\sigma^3}\sum_{i=1}^{n}\left(\alpha\left(\frac{x-\boldsymbol{v}_i}{\sigma}\right)(x-\boldsymbol{v}_i)^3\right)\right)$$

$$= \frac{x-\overline{\boldsymbol{v}}}{\sigma} + \frac{\sqrt{2\pi}}{\sigma^3}R(\sigma,x) + \left(\frac{1}{\sqrt{2\pi}}\left(\frac{x-\overline{\boldsymbol{v}}}{\sigma}\right) + \frac{1}{\sigma^3}R(\sigma,x)\right)^3$$

$$\cdot\beta\left(\frac{1}{\sqrt{2\pi}}\left(\frac{x-\overline{\boldsymbol{v}}}{\sigma}\right) + \frac{1}{\sigma^3}R(\sigma,x)\right)$$

$$= \frac{1}{\sigma}\left((x-\overline{\boldsymbol{v}}) + \frac{\sqrt{2\pi}}{\sigma^3}R(\sigma,x) + \left(\frac{1}{\sqrt{2\pi}}(x-\overline{\boldsymbol{v}}) + \frac{1}{\sigma^3}R(\sigma,x)\right)^3\right)$$

$$\cdot\beta\left(\frac{1}{\sqrt{2\pi}}(x-\overline{\boldsymbol{v}}) + \frac{1}{\sigma^3}R(\sigma,x)\right)$$

where $R(\sigma,x) := \frac{1}{n}\sum_{i=1}^{n}\alpha\left(\frac{x-\boldsymbol{v}_i}{\sigma}\right)(x-\boldsymbol{v}_i)^3$. As $\lim_{\sigma\to\infty}R(\sigma,x) = \frac{\alpha(0)}{n}\sum_{i=1}^{n}(x-\boldsymbol{v}_i)^3 < \infty$ and $\lim_{\beta\to\infty}\beta\left(\frac{1}{\sqrt{2\pi}}\left(\frac{x-\overline{\boldsymbol{v}}}{\sigma}\right) + \frac{1}{\sigma^3}R(\sigma,x)\right) = \beta(0) < \infty$, we obtain that

$$\lim_{\sigma\to\infty}\sigma\cdot\Phi^{-1}(cdf^*_{\boldsymbol{v},\sigma}(x)) = \lim_{\sigma\to\infty}\left((x-\overline{\boldsymbol{v}}) + \frac{\sqrt{2\pi}}{\sigma^2}R(\sigma,x) + \frac{1}{\sigma^2}\left(\frac{1}{\sqrt{2\pi}}(x-\overline{\boldsymbol{v}}) + \frac{1}{\sigma^2}R(\sigma,x)\right)\right)^3$$

$$\cdot\beta\left(\frac{1}{\sqrt{2\pi}}\left(\frac{x-\overline{\boldsymbol{v}}}{\sigma}\right) + \frac{1}{\sigma^3}R(\sigma,x)\right) = x-\overline{\boldsymbol{v}}.$$

$\square$

**Proposition 2.** *As the KDE kernel width $\sigma$ goes to zero, SQN approaches the behavior of classical quantile normalization. Given the notation introduced in Section 3, we have*

$$\lim_{\sigma \to 0} SQN_\sigma(x|\boldsymbol{v}) = CQN(x|\boldsymbol{v}) \tag{2}$$

*Proof.* With $\sigma$ approaching 0, the KDE kernels become infinitesimally thin until they are only present at their centers, which are their corresponding values. If there are multiple occurrences of a value in the input vector, their singular-valued kernels are added together, resulting in spikes of height proportional to the number of occurences of each value. The KDE integral, $cdf_{\boldsymbol{v},\sigma}^*$ consequently approaches a step function, mapping each value to the number of values in the vector that are less than or equal to it. This concludes the proof as the described behavior precisely matches the discrete rank mapping used in CQN. $\qquad\square$

