# OpenReview forum: "Structural Quantile Normalization: a general, differentiable feature scaling technique balancing gaussian approximation and structural preservation"
_ICLR.cc/2025/Conference — ICLR 2025 Conference Withdrawn Submission_

### Official Review · Reviewer_ybR2 · 2024-10-16

**Soundness:** 2
**Presentation:** 2
**Contribution:** 1
**Rating:** 3
**Confidence:** 5

**Summary:**

SQN is a feature preprocessing method that interpolates between z-score scaling and quantile normalization via the Gaussian KDE, utilizing PCHIP splines to achieve linear scalability while maintaining smoothness and monotonicity. This method is evaluated on the California housing dataset.

**Strengths:**

1. Feature preprocessing is an important task, as even deep learning methods such as TabPFN require preprocessing to deal with highly-skewed features.

2. Using PCHIP splines to achieve linear scalability while maintaining smoothness and monotonicity is an interesting proposal.

**Weaknesses:**

1. The experimental section is quite weak, only evaluating on the California housing dataset. The results would be much more convincing if the authors showed benefits on a benchmark of tabular datasets. And, while the authors claim that differentiability is valuable, but don't show that this provides any empirical benefits.

2. The authors evaluate their method in combination with feedforward neural networks, which are not even close to SotA for tabular data.

3. The originality of the proposal and the completeness of the related work section are seriously affected by the fact that the overall proposed approach is the same as KDIT (The Kernel Density Integral Transformation, McCarter, TMLR 2023). In both cases, one uses the KDE to smooth the pdf of a feature, then applies the inverse cdf of a reference distribution.

Granted, there are some differences, but these are minor:

(A) While the KDIT software package implements using the Gaussian as the reference output distribution, the KDIT paper only evaluates the uniform distribution as reference, thus interpolating between min-max scaling and quantile normalization, while SQN interpolates between z-score scaling and quantile normalization.

(B) KDIT uses the polynomial-exponential kernel (Fast exact evaluation of univariate kernel sums, Hofmeyr, TPAMI 2019) to obtain linear complexity, while SQN uses PCHIP splines for this. PCHIP splines are superior in that they enforce not just monotonicity, but also smoothness; whereas the KDIT is only almost-everywhere smooth. It's not clear that everywhere smoothness is really valuable, given that the authors combine their approach with neural networks with not-everywhere-smooth ReLU activations.

**Questions:**

With respect to weakness 2 above, I would suggest combining with either linear methods (which still have some value due to interpretability) or modern neural network approaches like TabPFN. Or perhaps the authors could evaluate their approach on neural network methods -- eg CSDI-T (Zheng & Charoenphakdee, TRL @ NeurIPS 2022) or TabDDPM (Kotelnikov et al, ICML 2023) -- for tabular data generation, which also require feature preprocessing.

---

> ### Author Response · Authors · 2024-11-26
>
> Dear Reviewer,
>
> Thank you for your time and effort in carefully reviewing our paper and for the constructive and valuable feedback. Your comments brought to our attention a closely related state-of-the-art method we overlooked. Based on this weakness we concluded withdrawing our paper. We also highly value your insightful suggestions on possible future directions.
>
> Thank you again, and warm regards.

---

### Official Review · Reviewer_5waL · 2024-10-29

**Soundness:** 1
**Presentation:** 2
**Contribution:** 1
**Rating:** 1
**Confidence:** 5

**Summary:**

The paper presents a new data-standardization method *applicable only to one dimensional data* which uses a specific procedure to monotonically perturb a KDE of a *one dimensional* applied to empirical measure, mapping it to a centered normal law.

Some extremely limited numerics on very a small dataset (used in basic textbooks and old Kaggle competitions) is used to evalued at the normalization procedure.

**Strengths:**

The authors provide a reasonable literature review and the graphs are very elegant.

**Weaknesses:**

There are no real justification of support of the method whatsoever.

- There is no theoretical support (there is a proposition in the appendix, which is formulated non-rigorously.  What time of convergence is this?  The KDE is a function of a random quantity, the empirical (random) measure associated with some one-dimensional law, then should I interpret this as $\omega$-wise convergence?

Therefore, this is no theoretical contribution.

- The numerics are not convincing, only a toy dataset (the California Housing Market) dataset is considered, which is something on sees in introductory text books.  Thus, there is also no experimental support for the proposed method.


---
Btw, the convergence in the proposition in the appendix is in the point-open topology, perhaps that should be mentioned (as the limit is difficult to interpret without reading the first line of the proof).  I think this should be clearly said in the statement.

**Questions:**

- The proposed method seemingly only works in one dimension.  In multi-dimensions, should one use Sklar's Theorem, if so, what do you do with Copulas?  If not, how does this apply to general probability measures on $\mathbb{R}^d$?

- What is the regularity (continuous, uniformly continuous, etc... for some reasonable metrization of the weak topology on a reasonable subset of $\mathcal{P}(\mathbb{R})$) of this operation in general?  Why should it be invertible in high-dimensions?

The most major question I have is: *Why use this method at all?* Specifically:

- Is there any *provable* mathematical theoretical guarantee that it *must* improve downstream learning?

- If you cannot provide a guarantee, then there must at least be convincing experimental evidence (what is offered certainly is not).

- What is the take-way of Propostion 1, why is it important, and what is the concrete implication on the proposed method?

Even if the above questions could be answered, I must ask: how does this work in high dimensions?  This is not obvious to me, I'm assuming it is not to the authors either or the method would have been presented in greater generality.

---

> ### Author Response · Authors · 2024-11-25
>
> Dear Reviewer,
>
> We have decided to withdraw our paper in light of the similar work that Reviewer ybR2 brought to our attention. Thank you for taking the time to review our work. We found several misunderstandings in your review which we address below. Please also find our responses to your questions.
>
> *Like all scaling methods, our method extends to multiple dimensions by applying the transformation feature-wise. This approach is straightforward, standard practice in ML, and doesn't require additional complex mathematics.
>
> *The transformation is indeed invertible. The PCHIP algorithm produces a cubic spline, which can be inverted straightforwardly using the cubic formula.
>
> *The KDE is a function of a discrete feature vector, not a random quantity. We cannot see where this impression comes from. The convergence in the proposition refers to standard convergence. Reviewer 4 pointed out a similar proof in a closely related method (that we overlooked) that was published at a top conference last year.
>
> *It's uncommon for a technique in CS to be universally better than all others in a mathematically provable way. Different approaches tend to work better for different applications, which adds to the richness of the field.
>
> *Using a basic dataset for proof of concept evaluation does not necessarily take away from the quality of the evaluation. We would appreciate a more detailed critique of the evaluation procedure.
>
> *The main takeaway of Proposition 1 is that our method interpolates between standardization (z-score scaling) and quantile normalization.
>
> Thank you,

---

### Official Review · Reviewer_4HnD · 2024-11-03

**Soundness:** 3
**Presentation:** 3
**Contribution:** 2
**Rating:** 3
**Confidence:** 4

**Summary:**

The paper proposed a feature scaling method that transforms a feature such that the feature is close to normally distributed after the transformation. A kernel density estimation is fitted to model the pdf of the feature, then the inverse Gaussian cdf is applied to the cdf estimate of the feature. The kernel density estimation allows the method to keep the local structure of the original distribution. Experiments in one data set demonstrate that the proposed methods outperform other scaling methods.

**Strengths:**

The proposed methods are well-motivated and explained well. It covers existing methods as special cases and can be computed efficiently with the proposed FAST-SQN version.

**Weaknesses:**

Empirical validation is not enough

1. Only one real data experiment is provided
2. It is hard to tell if the performance improvements are significant, e.g. in table 1, model 3, Fast-SQN only improves RMSE by 0.001 over YJN, mean and standard deviation over multiple runs should be reported.
3. The proposed method should be a general scaling method for any feature, but it seems only the target feature is normalized in the experiments (line 390 in the paper). I thought rescale input features would be more useful in model training.
4. One advantage that is emphasized for the proposed method is its differentiability, but if it is only used in the experiment setting in the paper, differentiability doesn't seem to have any advantage.

Overall, I think the empirical validation is far from convincing. The proposed methods should be able to apply to a wide variety of data. Much more experiments are needed to really demonstrate that it is superior to existing methods. If it can indeed improve model performance across a lot of data sets and especially if it can be applied similarly to batch-norm or layer-norm layer, this could be a major contribution to the community, but the current results don't support the claim.

**Questions:**

See weakness

---

> ### Author Response · Authors · 2024-11-25
>
> Dear Reviewer,
>
> We have decided to withdraw our paper in light of the similar work that Reviewer ybR2  brought to our attention. We would like to thank you for your valuable and constructive feedback, and we wish to address two of the specific weaknesses you raised:
>
> *Weakness 3: We agree. Transforming all features during testing would likely have yielded more interesting and meaningful results.
>
> *Weakness 4: While it is true that the differentiability of our method offers no real advantage in a standard preprocessing setting, we envisioned its application as a model-internal normalization component—similar to Layer Normalization in transformers—where differentiability is required for trainability, as we pointed out in the introduction and conclusion of the paper.
>
> We will make sure to clarify these points in a future version of the paper.
>
> Best regards and thank you again for the constructive feedback,

---

### Official Review · Reviewer_vZQj · 2024-11-03

**Soundness:** 3
**Presentation:** 3
**Contribution:** 2
**Rating:** 3
**Confidence:** 3

**Summary:**

The paper introduces a normalisation scheme called Structural Quantile Normalisation (SQN). Unlike many other methods, the proposed transformation is differentiable. The authors blend in the ideas of quantile normalisation, kernel density estimation, and PCHIP to propose their method. The method can be slow in its true form, but the authors provide a modification called Fast-SQN to counter it.

**Strengths:**

1. The paper is well-written and easy to follow.
2. The experiments in Table 1 show the efficacy of the method on the given dataset.
3. Fast-SQN solves the computational efficiency issue by cleverly adding a spline interpolation layer.

**Weaknesses:**

1. Table 1 shows better performance of the proposed method but lacks the error bounds, it might be helpful to see the fluctuation if different random seeds are employed for the same model.
2. Authors cited that Gaussianisation techniques can mean losing intrinsic patterns in the data and are non-differentiable. However, I would argue that methods like Normalising Flows are differentiable, and a few such layers in the beginning might help with this. It would be interesting to see if NF-transformed data lags in performance compared to the methods reported in Table 1.
3. Only one dataset is discussed, it is hard to say if the results might hold for any kind of data.
4. Can this transformation also help with Image data? or use of KDE limits applicability to high dimensionality?
5. Can using these transformation layers with methods that aim at transforming the distribution to standard normal can we reduce the computational complexity of those methods?

**Questions:**

Please see weaknesses

---

> ### Author Response · Authors · 2024-11-25
>
> Dear Reviewer,
>
> We have decided to withdraw our paper in light of the similar work that Reviewer ybR2 brought to our attention. We would like to thank you for your valuable and constructive feedback, and we wish to address two of the specific weaknesses you raised:
>
> *Weakness 4: Since our method transforms training data feature-wise, there should not be a problem when extending it to higher dimensions.
>
> *Weakness 5: Our method does not aim to transform data into a precise normal distribution due to the intrinsic information loss that such a transformation would entail. Instead, it seeks to balance achieving semi-Gaussian properties while preserving the "local structure" of the data.
>
> We will make sure to clarify these points in a future version of the paper.
>
> Best regards and thank you again for the constructive feedback,

---

### Note · Authors · 2024-11-26

I have read and agree with the venue's withdrawal policy on behalf of myself and my co-authors.